# GRAPH-GUIDED SPARSE LEARNING VIA BOOLEAN RELAXATION

## ABSTRACT

We propose a novel graph-guided sparse learning model using $\ell_0$ norm. The proposed model addresses a key limitation of existing methods, which enforce neighboring variables to have similar coefficients. We introduce a novel relaxation for the proposed problem. Our approach is based on reformulating the original model exactly as a Boolean convex program. We analyze the first-order relaxation and derive the necessary and sufficient conditions for exactness. We further show that these conditions are satisfied with high probability on random ensembles. Unlike existing methods, our relaxations provide lower bounds on the objective and can be verified whether the relaxation is exact. When the relaxation is not exact, we show that a rounding scheme based on the relaxed solutions leads to provably good feasible solutions. We numerically illustrate the outperformance of our novel relaxation in both simulation data and the real-world gene regulation inference task, demonstrating significant improvement of the proposed model.

## 1 INTRODUCTION

In many modern data analysis tasks, particularly those involving high-dimensional datasets, it is crucial to identify the most relevant features while preserving model interpretability and predictive accuracy. Sparse learning methods Tibshirani (1996); Zou & Hastie (2005) address this challenge by promoting solutions with only a small number of active (non-zero) coefficients, effectively filtering out unimportant variables. However, in many applications, the features exhibit intricate relationships or dependencies—such as spatial Li & Li (2008); Huang et al. (2008), temporal Tibshirani et al. (2005); Zou & Hastie (2005), or functional connections Zou & Hastie (2005); Yuan & Lin (2006)—that standard sparse techniques fail to capture. This need for modeling inter-feature structure leads naturally to graph-guided sparse learning, where domain knowledge is encoded as a graph and used to guide the feature selection or coefficient estimation process Huang et al. (2009); Chen et al. (2012); Hallac et al. (2015).

In this paper, we propose a novel graph-guided sparse learning model formulated as:

$$P^* = \min_{\|w\|_0 \le k} \left\{ F(w) = \sum_{i=1}^n f(w^\top x_i, y_i) + \frac{1}{2}\rho\|w\|_2^2 + \mu\Phi_G(w) \right\}, \tag{1}$$

where $f(\cdot)$ is a loss function, $\{(x_i, y_i)\}_{i=1}^n$ denotes a set of training samples with observation $x_i \in \mathbb{R}^d$ ($d$ variables) and response $y_i \in \mathbb{R}$, and $w \in \mathbb{R}^d$ is the model parameter. The term $\|w\|_0$ in the constraint $\|w\|_0 \le k$ counts the number of non-zero entries in $w$, promoting unstructured sparsity.

To incorporate structural information among variables, we define the graph-guided regularization $\Phi_G(w)$ in equation 1 as:

$$\Phi_G(w) = \sum_{(i,j)\in E} H_{ij}\|w_i\|_0\|w_j\|_0 = u^\top Hu, \tag{2}$$

where $G = (V, E)$ is a graph with nodes $V$ representing variables and edges $E$ encodes inter-variable relationships between $d$ variables. The vector $u = [\|w_1\|_0, \|w_2\|_0 \ldots, \|w_d\|_0]^\top$ capture the sparsity pattern of $w$. The matrix $H$ in equation 2 is a positive semidefinite matrix, which represents the topological properties encoded in $G$ and can take various forms depending on the application, including but not limited to:

- Laplacian Matrix: $H = L$. The objective of using $H = L$ is to identify a sparse support $\mathrm{supp}(w) = \{j \in \{1, \ldots, d\} \mid w_j \neq 0\}$ that lies on a minimum cut of the graph $G$;

- Normalized Laplacian Matrix: $H = D^{-1/2}LD^{-1/2}$. The objective of using $H = D^{-1/2}LD^{-1/2}$ is to identify a sparse support $\mathrm{supp}(w)$ that lies on a normalized cut of the graph $G$;

- Gram Matrix: $H$ can be any Gram matrix that encodes the pairwise similarities between variables. In the context of graph-guided learning, $H$ can be interpreted as the adjacency matrix of a graph.

Here, $A \in \mathbb{R}^{d \times d}$ is the adjacency matrix of $G = (V, E)$, $D$ is the diagonal degree matrix $D$ with $D_{ii} = \sum_j A_{ij}$, and $L = D - A$ is the graph Laplacian matrix. Notably, $H$ is required to be positive semidefinite in the proposed model equation 1. The graph $G = (V, E)$ is typically predefined based on domain knowledge and may not necessarily depend on the observed data $x_i$.

## 1.1 RELATED WORKS

There are two main classes of graph-guided sparse learning models related to our proposed model equation 1. The first incorporates graph structure through regularization Tibshirani et al. (2005); Tibshirani (2011); Hallac et al. (2015); Li & Li (2010a). The generalized fused lasso (GFL) Tibshirani et al. (2005) and generalized lasso Tibshirani (2011) introduces penalties of the form $\lambda_1 \|w\|_1 + \lambda_2 \sum_{(i,j) \in E} \|w_i - w_j\|_1$, applying $\ell_1$ norm to both individual coefficients and their pairwise differences. Various algorithms have been proposed to solve GFL, including proximal-gradient algorithms Xin et al. (2014) and block coordinate-descent algorithms Wainwright (2019); Ohishi et al. (2022). Network lasso Hallac et al. (2015), which applies similar penalties as GFL but omits the $\ell_1$-term on the individual coefficients, can be solved by the Alternating Direction Method of Multipliers (ADMM) Boyd et al. (2011); Hallac et al. (2015); Tansey & Scott (2015); Zhu (2017); Cao et al. (2018); Yu et al. (2025). Adaptive Grace models Li & Li (2010a) applies $\lambda_1 \|w\|_1 + \lambda_2 \sum_{(i,j) \in E} \|w_i - w_j\|_2^2$ regularization, to promote smoothness over the graph. However, these existing models tend to enforce coefficient similarity among neighboring variables, even when the underlying coefficients differ significantly or have opposite signs, thus limiting their flexibility.

The second line of work enforces graph structure via a convex constraint on the support Needell & Tropp (2009); Baraniuk et al. (2010); Hegde et al. (2015a;b; 2016); Locatello et al. (2018); Zhou et al. (2019); Zhou & Sun (2022). Representative algorithms include gen_mp Locatello et al. (2018), cosamp Needell & Tropp (2009), graph_cosamp Hegde et al. (2015b), graph_iht Hegde et al. (2016), and dmo_acc_fw Zhou & Sun (2022). All solve

$$\min_{w \in \mathcal{D}(C, \mathcal{M})} f(w), \qquad \mathcal{D}(C, \mathcal{M}) = \mathrm{conv}\{w \in \mathbb{R}^d : \mathrm{supp}(w) \in \mathcal{M}(G, s, g), \|w\|_2 \leq C\},$$

where $\mathcal{M}(G, s, g) = \{S \subseteq V : S = S_1 \cup \cdots \cup S_g, |S| \leq s\}$, with each $S_i$ a connected subgraph of $G = (V, E)$. Although this guarantees the selected indices form at most $g$ connected components, it ignores (i) the internal connectivity of each component and (ii) the connectivity between selected and unselected nodes—two graph properties that matter when training on noisy samples.

We extend the analytical framework in Wang et al. (2023) to incorporate the term $\Phi_G(w)$ in equation 1 and derive our own theorems established in this paper.

## 1.2 A MOTIVATING EXAMPLE

To demonstrate the advantage of the proposed model equation 1 over existing models, we construct a synthetic dataset where $x_i \in \mathbb{R}^{12}$ is sampled from a standard multivariate normal distribution $\mathcal{N}(\mathbf{0}, I)$ and $y_i$ is generated by $y_i = x_i^\top w$, where $w$ is the ground-truth coefficients for the variables as shown in Fig. 1b. The predefined unweighted graph $G = (V, E)$ is given by Fig. 1a. The goal of the test is to compare different models on their ability to recover the sparse solution that consists of variables from 1 to 5.

We use the Laplacian matrix $H = L$ for equation 2 in our model equation 1. After some derivation (omitted for brevity), we find that $\Phi_G(w)$ in equation 2 can be written as $\Phi_G(w) = \sum_{(i,j) \in E} (\|w_i\|_0 - \|w_j\|_0)^2$ when $H$ is the Laplacian matrix. Compared to graph regularization $\sum_{(i,j) \in E} \|w_i - w_j\|_1$ in

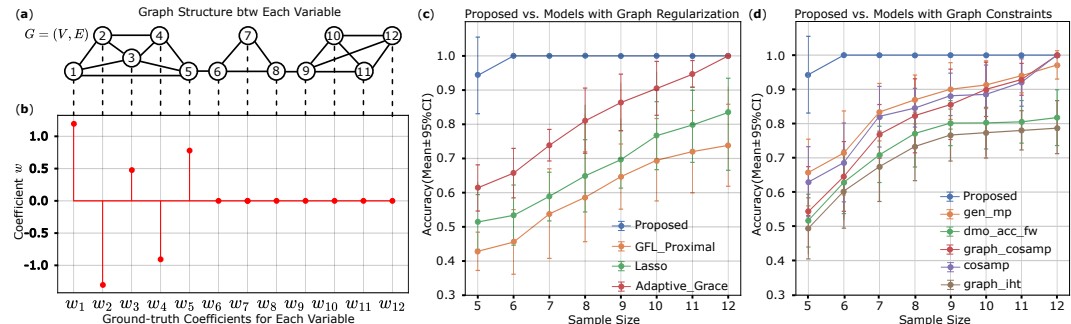

Figure 1: (a) The graph $G = (V, E)$ of the variables. (b) The ground-truth coefficients for 12 variables. (c) Comparison between the proposed model and models using graph regulation over 10 runs in terms of recovery accuracy. (c) Comparison between the proposed model and models based on graph-structured constraints over 10 runs in terms of recovery accuracy.

GFL and graph regularization $\sum_{(i,j) \in E} \|w_i - w_j\|_2^2$ in Adpative Grace, our proposed regularization $\sum_{(i,j) \in E} (\|w_i\|_0 - \|w_j\|_0)^2$ does not affect the actual magnitudes of $w_i$ and $w_j$, focusing instead on the structure of the support. Such differences can easily be demonstrated when considering $w_1 = 1.2$ and $w_2 = -1.8$ as shown in Fig. 1a-b.

When $H$ is the Laplacian matrix, $\Phi_G(w)$ in equation 2 is also equal to $\Phi_G(w) = u^\top L u$, which encourages the support of the sparse solution to align with a minimum cut of $G$. In contrast, prior models based on graph-structured constraints Needell & Tropp (2009); Baraniuk et al. (2010); Hegde et al. (2015a;b; 2016); Locatello et al. (2018); Zhou et al. (2019); Zhou & Sun (2022)focus on selecting variables from connected components in $G$, but do not prioritize minimum cut solutions.

We compare our model equation 1 against two types of baselines: (1) sparse models with graph regularization, including GFL_Proximal Xin et al. (2014) and Adaptive Grace Li & Li (2010a), as shown in Fig. 1c, and (2) convex optimization methods over graph-structured constraints (gen_mp Locatello et al. (2018), cosamp Needell & Tropp (2009), graph_cosamp Hegde et al. (2015b), graph_iht Hegde et al. (2016), and dmo_acc_fw Zhou & Sun (2022)), as shown in Fig. 1d. For each model, hyperparameters are selected via cross-validation. We assume the number of selected variables is known and fixed at 5, and recovery accuracy is used as the evaluation metric. The recovery accuracy is defined as the proportion of ground-truth variables that are correctly identified among the 5 variables returned by the method.

As shown in Fig. 1c-d, the proposed method consistently outperforms existing baselines in recovering the ground-truth support and achieving higher accuracy across sample sizes, especially when the sample size is small.

### 1.3 CONTRIBUTIONS

To fill the gap, in this paper, we propose to solve $P^*$ equation 1. We first reformulate $P^*$ as a Boolean convex program. We further establish an analytical and algorithmic framework for the Boolean relaxation of $P^*$, which includes a theorem stating the equivalent condition for the relaxation to achieve the exactness (i.e., the optimal integral solution) and a rounding scheme that produces an integral solution when the optimal relaxation solution is fractional. We demonstrate the power of our equivalent condition theorem by applying it to an ensemble of random problem instances that is challenging and popularly used in literature and proving that our Boolean relaxation of $P^*$ achieves exactness with high probability and nearly optimal sample complexity. Our **contributions** are threefold:

1. **Model Innovation**: We propose a novel formulation $P^*$ equation 1 that incorporates graph structure into the sparse learning model using $\ell_0$ norm. Unlike prior models using graph regularization Tibshirani et al. (2005); Tibshirani (2011); Hallac et al. (2015); Li & Li (2010a)—which encourage coefficient similarity among adjacent variables—our model equation 1 allows the learned coefficients to differ more freely, avoiding unnecessary smoothing imposed by graph regularization. Comparing prior models based on graph-structured constraints Needell & Tropp (2009); Baraniuk et al. (2010); Hegde et al. (2015a;b; 2016);

Locatello et al. (2018); Zhou et al. (2019); Zhou & Sun (2022)—which overlook both the internal and external connectivity of selected variables on the graph, our model equation 1 is able to capture such topological structure through the choice of $H$ in equation 2.

2. **Theoretical Advancement**: We provide an exact reformulation of $P^*$ equation 1 as a Boolean convex program. We apply Boolean relaxation to reformulation and prove the relaxation is tight and can achieve the exactness with high probability and the nearly optimal sample complexity for ensembles of random problem instances.

3. **Empirical Validation**: We conduct extensive experiments to demonstrate that our framework significantly outperforms the state-of-the-art methods **when the sample size is small** on simulated datasets. Furthermore, we show the outperformance of our framework in a real-world application: gene regulation inference.

## 2 BOOLEAN RELAXATION FOR GRAPH-GUIDED SPARSE LEARNING

In this section, we describe how Boolean relaxation is used to solve the proposed model $P^*$ equation 1 with a theoretical tightness guarantee. The organization of this section is as follows. In Section 2.1, we introduce the original problem and present its exact Boolean formulation. In Section 2.2, we propose a relaxed Boolean program and establish a condition under which the relaxation is guaranteed to yield an integral (Boolean) solution, thus ensuring tightness. Finally, in Section 2.3, we outline a rounding strategy to be applied in cases where the relaxed program does not produce integral solutions.

### 2.1 GRAPH-GUIDED SPARSE LEARNING AND ITS FORMULATION VIA BOOLEAN CONSTRAINTS

We consider the learning problem defined in (1). In the following theorem, we show that the problem can be reformulated as a convex program with additional Boolean variables and constraints, which will naturally lead to the convex Boolean relaxation algorithm in the later sections

**Theorem 2.1.** *Suppose that the function $t \mapsto f(t; y)$ is closed and convex for each $y \in \mathcal{Y}$. The Legendre-Fenchel conjugate of $f$ is $f^*(s; y) := \sup_{t \in \mathbb{R}} \{st - f(t; y)\}$. Then for any $\rho > 0$, the structured sparse learning problem $P^*$ can be represented by the following Boolean program:*

$$\min_{u \in \Gamma} \left\{ \underbrace{\max_{v \in \mathbb{R}^n} \left[ -\frac{1}{2\rho} v^\top X D(u) X^\top v - \sum_{i=1}^n f^*(v_i, y_i) \right]}_{G(u)} + \mu \ \mathrm{tr}(u^\top H u) \right\} \tag{3}$$

*where $D(u) := \mathrm{diag}(u)$ is a diagonal matrix with Boolean variables $u \in \mathbb{R}^d$ on its diagonal, the matrix $H$ is a specified positive semidefinite matrix, and $\Gamma$ is the constraint set for $u$:*

$$\Gamma = \left\{ u \ \middle| \ u \in \{0, 1\}^d; \quad \sum_{i=1}^d u_i \leq k \right\}$$

The proof of Theorem 2.1 can be found in the supplementary materials Section A. In the statement, $u$ is a vector of the Boolean indicators for the supports of the individual features. $G(u)$ in equation (3) is convex in $u$ because it is the maximum of a family of functions that are linear with $u$. The second part is also convex in $u$ because the $H$ is positive semi-definite by assumption.

However, the overall problem remains computationally challenging due to the Boolean constraint $u \in \{0, 1\}^d$. In the following subsection, we relax this constraint to obtain a convex program that can be efficiently solved for many commonly used loss functions $f$.

## 2.2 Convex Program through Boolean Relaxation and Theoretical Conditions for Exactness

Apply interval relaxation to Boolean vector variables $u_i$, we obtain the Boolean relaxation for $P^*$:

$$P_{\text{BR}} = \min_{u \in \Omega} \left\{ \max_{v \in \mathbb{R}^n} \left[ -\frac{1}{2\rho} v^\top X D(u) X^\top v - \sum_{i=1}^n f^*(v_i, y_i) \right] + \mu \, \text{tr}(u^\top H u) \right\} \tag{4}$$

where

$$\Omega = \left\{ u \,\middle|\, u \in [0,1]^d; \quad \sum_{i=1}^d u_i \leq k \right\}$$

$P_{\text{BR}}$ is a convex program and can be solved by the sub-gradient-based optimization algorithm Nesterov (2009) if the inner maximization problem can be solved efficiently. In general, $P_{\text{BR}}$ can also be converted into a minimax optimization problem and solved by methods in Lin et al. (2020).

We now determine when $P_{\text{BR}}$ achieves the exact solution of $P^*$. The following theorem (proved in the supplementary materials Section B) provides the equivalent condition for exactness.

**Theorem 2.2.** *Suppose each feature belongs to at most one cluster and that the optimal integral solution $\hat{u} = (\hat{u}_i)$ for $P^*$ selects exactly $k$ features. Then, the optimal solution of $P_{\text{BR}}$ also recovers $\hat{u}$ if and only if there exist non-negative values $\lambda$ such that:*

$$\hat{v} \in \arg\max_{v \in \mathbb{R}^n} \left\{ -\frac{1}{2\rho} v^\top X D(u) X^\top v - \sum_{i=1}^n f^*(v_i, y_i) \right\}.$$

*For $i \in I_n$, it holds that*

$$\frac{1}{2\rho} \left( X_i^\top \hat{v} \right)^2 - \mu (Hu)_i \leq \lambda;$$

*For $i \in I_s$, it holds that*

$$\frac{1}{2\rho} \left( X_i^\top \hat{v} \right)^2 - \mu (Hu)_i \geq \lambda.$$

*Here, $I_s$ denotes the set of indices for the selected features, $I_n$ denotes the set of indices for the features that are not selected, and $X_i$ denotes the $i$-th column of the design matrix $X$.*

**The Special Case of Least-Squares Regression**   Among various loss functions, the squared loss $f(t; y) = \frac{1}{2}(t - y)^2$ stands out as particularly significant for least-squares regression due to its analytical tractability and widespread practical adoption. The Legendre-Fenchel conjugate of this loss function is given by $f^*(s; y) = \frac{s^2}{2} + sy$. Substituting this conjugate into our general framework yields the following specialized convex relaxation:

$$P_{\text{BR}} = \min_{u \in \Omega} \left\{ \frac{1}{2} y^\top \left( \rho^{-1} X D(u) X^\top + I_n \right)^{-1} y + \mu \cdot \text{tr}(u^\top H u) \right\} \tag{5}$$

where $D(u) := \text{diag}(u)$ is the diagonal matrix with $u$ on its diagonal.

Let $S = \text{supp}(\hat{u})$ denote the support of the unique optimal solution and define $B := \left( I_n + \rho^{-1} X_S X_S^\top \right)^{-1}$. For least-squares regression, Theorem 2.1 specializes to:

**Corollary 2.3** (Exact Recovery for Least-Squares). *Under cluster exclusivity (each feature belongs to at most one cluster) and exact $k$-sparsity of the optimal integral solution $\hat{u}$, the relaxation $P_{\text{BR}}$ recovers $\hat{u}$ if and only if there exists $\lambda \geq 0$ such that:*

$$\frac{1}{2\rho} (X_i^\top B y)^2 - \mu (Hu)_i \leq \lambda, \quad \forall i \in I_n,$$

$$\frac{1}{2\rho} (X_i^\top B y)^2 - \mu (Hu)_i \geq \lambda, \quad \forall i \in I_s.$$

*where $I_s$ and $I_n$ denote the sets of selected and non-selected features as in Theorem 2.2.*

## 2.3 RANDOMIZED ROUNDING WITH PROVABLE GUARANTEES

When the Boolean relaxation yields fractional solutions ($\bar{u} \in [0,1]^d \setminus \{0,1\}^d$), we employ randomized rounding Pilanci et al. (2015); Wang et al. (2023) to recover feasible integral solutions. This probabilistic technique preserves expectation while maintaining approximation guarantees. Given fractional solution $\bar{u}$, generate $u \in \{0,1\}^d$ via $u_i \sim \text{Bernoulli}(\bar{u}_i)$, $\forall i \in [d]$ with independent trials across coordinates.

It is straightforward to verify that the Boolean solution generated by this method matches the fractional solution in expectation, that is, $\mathbb{E}[u] = \bar{u}$, Moreover, the expected $\ell_0$-norms of the solutions are given by $\mathbb{E}[\|u\|_0] = \sum_{i=1}^{d} \Pr[u_i = 1] = \sum_{i=1}^{d} \bar{u}_i \leq k$.

Using these expectation bounds, we invoke standard concentration inequalities to prove that the rounded Boolean solution is sparse and nearly optimal with high probability.

**Theorem 2.4** (Optimality gap). *Let $\bar{u}$ be the optimal solution of the relaxed problem and let $u$ be its rounded integral counterpart. For any $\delta > 0$, with probability at least*

$$1 - c_1 \exp(-c_2 k \delta^2) - \min(r,n)^{-c_3} - 2\exp\left(-c_4 \min\left\{\frac{\log n}{\|H\|_F^2}, \frac{\sqrt{\log n}}{\|H\|}\right\}\right) - 2\exp\left(-c_5 \frac{\log n}{\|H^\top \bar{u}\|_2^2}\right),$$

*the vector $u$ satisfies $\|u\|_0 \leq (1+\delta)k$ and*

$$G(u) - P^* \leq O\left(\rho^{-1}\sqrt{r \log\min(r,n)} + \mu\sqrt{\log n}\right),$$

*where $r$ is the number of fractional coordinates in $\bar{u}$, $\|H\|$ is the operator norm of $H$, and $\|H\|_F$ its Frobenius norm.*

The proof is provided in Section C of the supplementary material.

As $n$ grows, every exponential tail term decays, so the success probability approaches 1. The gap becomes negligible when either $\mu$ is small, the number of fractional solutions $r$ is modest, or $\rho$ is large (strong $\ell_2$ regularization).

Once the integral support $u$ is obtained, the weight vector for the original problem, Eq. equation 1, is $w := \arg\min_w F(D(u)w)$, i.e. we solve the loss restricted to the selected coordinates.

## 3 THEORETICAL GUARANTEES OF $P_{\text{BR}}$ ON ENSEMBLES OF RANDOM INSTANCES

In this section we invoke Corollary 2.3 to prove that the relaxed program is tight—that is, it recovers the integral optimum with high probability—on the ensembles of random instances. We restrict attention to least-squares regression and take $H$ to be the graph Laplacian $L = D - A$.

The synthetic data generation framework comprises four key components: (1) a design matrix $X \in \mathbb{R}^{n \times d}$, (2) a graph representation $G$ (expressed through adjacency matrix $A$ or Laplacian $L$), (3) a regression weight vector $w \in \mathbb{R}^d$, and (4) a response vector $y \in \mathbb{R}^n$.

Given sample size $n$ and feature dimension $d$, we construct the design matrix $X$ with independent and identically distributed (i.i.d.) entries drawn from the standard normal distribution $\mathcal{N}(0,1)$. For feature selection, we assume without loss of generality that the first $k$ features form the relevant cluster with nonzero contributions to the response variable, while the remaining $d - k$ features constitute the irrelevant cluster. This permutation invariance follows from the symmetric structure of our random ensemble.

Select $k$ features, set their coefficients to random signs $\pm 1/\sqrt{k}$, and set all others to 0 so that $\|w\|_2 = 1$. Construct a random graph where any two features of the same status (both selected *or* both non-selected) are connected with probability $p > 1/2$, while a mixed pair (one selected, one non-selected) is connected with probability $q < 1/2$. This models inperfect and noisy prior connectivity.

The response vector follows the standard linear model:

$$y = Xw + \epsilon, \quad \epsilon \sim \mathcal{N}(0, \gamma^2 I_n)$$

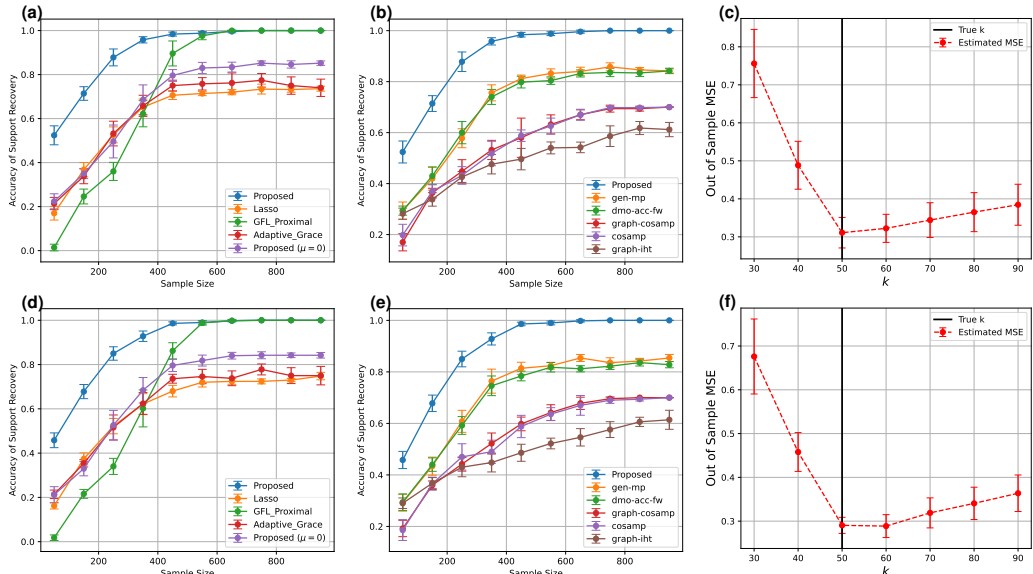

Figure 2: (a)–(c) show results for **Random Ensemble I** with $p = 0.9$ and $q = 0.2$. (a) compares support recovery performance between the proposed method and general fused lasso baselines; (b) compares against graph-constrained methods; (c) reports out-of-sample mean squared error (MSE) for different values of $k$. (d)–(f) show corresponding results for $p = 0.7$ and $q = 0.2$. All results are averaged over 10 random instances, with error bars indicating 95% confidence intervals.

where the additive Gaussian noise has signal-to-noise ratio $\text{SNR} = 1/\gamma^2$. Our objectives are twofold: exact support recovery of $w$ and accurate coefficient estimation under this synthetic data regime.

**Theorem 3.1.** *Consider the random ensemble with parameters* $(n, d, k, \gamma, p, q)$ *and observed response* $y = Xw + \epsilon$. *Let* $\rho = n^{1/2+\delta}$ *for some* $\delta > 0$ *such that* $\rho \geq 6.8 \cdot k(S + N)$, *where* $S$ *is the max irrelevant-to-relevant edges and* $N$ *is max relevant-to-irrelevant edges:*

$$S = \max_{i \in I_n} |\{j \in I_s : j \sim i\}|, \quad N = \max_{i \in I_s} |\{j \in I_n : i \sim j\}|$$

*where* $I_s$ *denotes the set of indices for the selected features,* $I_n$ *denotes the set of indices for the features that are not selected. Then with probability exceeding*

$$1 - d \exp\left(-\Omega\left(\frac{n^{2\delta}}{\gamma^2 k}\right)\right) - d \exp\left(-\Omega(n^{1-2\delta})\right),$$

*the convex relaxation* $P_{\text{BR}}$ *(5) with* $H = L$ *admits a unique optimal solution* $u^*$ *that exactly corresponds to the ground truth feature selection, i.e.,* $u^* = \mathbf{1}_{\{w_i \neq 0\}}$.

The proof is given in Section D of the supplementary material. An analogous result holds if $H$ is the normalized Laplacian $\widetilde{L}$; see Section E.

Choosing $\delta = \frac{1}{4}$ makes both exponents $n^{1/2}$, so the probability becomes $1 - d \exp[-\Omega(n^{1/2}/(\gamma^2 k))]$. Thus $n \gtrsim (\gamma^2 k)^2 \log^2 d$ ensures recovery with probability $1 - o(1)$. Taking $\delta = \frac{1}{2} - \frac{\log \log n}{\log n}$ drives the first term up to $d \exp(-n/(\gamma^2 k(\log n)^2))$ while the second is $d \exp(-(\log n)^2)$. Then $n = \omega((k/\gamma^2) \log d)$ is enough for $1 - o(1)$ success, matching the information-theoretic lower bound of Wainwright (2009) up to log factors.

## 4 EXPERIMENTS

In this section, we conduct extensive experiments to evaluate the effectiveness of our proposed sparse selection models under the $\ell_0$-regularized least-squares regression framework. We assess performance on both synthetic datasets and a real-world application in gene regulation inference. Our method is benchmarked against state-of-the-art techniques and analyzed in terms of support recovery accuracy, robustness to noise.

To measure support recovery accuracy, we define the metric $A_I(w)$ as

$$A_I(w) := \frac{|\operatorname{supp}(w) \cap \operatorname{supp}(w^{\text{true}})|}{|\operatorname{supp}(w^{\text{true}})|},$$

where $\operatorname{supp}(w) = \{i : w_i \neq 0\}$ denotes the support of weight vector $w$, and $w^{\text{true}}$ represents the ground-truth coefficients.

We compare our proposed model against two types of baselines: (1) sparse models with graph regularization, including Adaptive_Grace Li & Li (2010b) and GFL_Proximal Xin et al. (2014); (2) convex optimization methods over graph-structured constraints, including gen_mp Locatello et al. (2018), cosamp Needell & Tropp (2009), graph_cosamp Hegde et al. (2015b), graph_iht Hegde et al. (2016), and dmo_acc_fw Zhou & Sun (2022).

We also include Lasso Tibshirani (1996) and Boolean Lasso Pilanci et al. (2015). Note that Boolean Lasso is a special case of our method with $\mu = 0$, referred to as Proposed ($\mu = 0$), and serves as an ablation study for the graph regularization component.

We solve the proposed $\ell_0$-constrained model equation 5 using a projected quasi-Newton (PQN) method Schmidt et al. (2009) with Armijo line search. Implementation details are provided in Section F of the supplement. Time complexity analysis ($\mathcal{O}(\min(n, d)^3)$) and runtime comparisons are included in Section G. To ensure scalability, the main computational cost lies in the projection step within PQN, which can be handled efficiently using commercial solvers or fast projection methods such as Ang et al. (2021), given that the convex constraints remain simple even in large-scale settings. All experiments were run on an Apple M2 Max MacBook Pro (12-core CPU, 38-core GPU, 96GB RAM) using Python 3.11. Timing results exclude data loading and preprocessing.

## 4.1 RANDOM ENSEMBLE I: CORRELATION IN DESIGN MATRIX

We begin by evaluating Random Ensemble I using the synthetic setup described in Section 3, with $d = 1000$, $k = 50$, and $\gamma = 0.5$, leading to a signal-to-noise ratio (SNR) of approximately 4. For graph construction, we consider two parameter settings, $(p = 0.9, q = 0.2)$ and $(p = 0.7, q = 0.2)$. In the first setting, more correct structure is retained, whereas both settings introduce additional noisy edges.

To make the prediction task more challenging and to demonstrate robustness, we introduce correlation between 30% of randomly selected pairs of selected and non-selected features in the design matrix. The goal is to recover the ground-truth weight vector $w^{\text{true}}$, which contains $k = 50$ contributing features. All hyperparameters—except $k$—are chosen via 5-fold cross-validation based on mean squared error (MSE). Since Lasso and GFL_Proximal do not always return exactly $k$ nonzero coefficients, we tune their regularization parameters and, when needed, retain only the top $k$ features with the largest estimated weights in magnitude.

For convex optimization methods over graph-structured constraints, we set hyperparameters according to the authors' recommendations. One important modeling choice is to set the number of connected components $g = 1$, consistent with our setup. Specifically, the subgraph induced by the selected features follows an Erdős–Rényi model $G(50, p)$. According to Theorem 7.3 in Bollobás & Bollobás (1998), when $p > \log(50)/50 \approx 0.08$, the graph is connected with high probability. Thus, assuming a single component is well justified in this setting.

Fig. 2 (a), (b), (d), and (e) show that as the sample size $n$ increases, support recovery accuracy improves across all methods. However, only the proposed method and GFL_Proximal converge to 1 when leveraging graph information, while the others plateau around 0.8, indicating a lack of robustness to feature correlation. In contrast, our method consistently achieves exact recovery and converges faster than GFL_Proximal, especially in low-sample regimes where noise and correlation are more pronounced.

In practice, the true number of contributing features $k$ is often unknown. A standard strategy to estimate this is through cross-validation using out-of-sample Mean Squared Error (MSE) as the selection criterion. We evaluate whether our method can correctly identify the true sparsity level $k$ by analyzing the out-of-sample MSE. As shown in Figs. 2(c) and 2(f), our method achieves the lowest MSE when the candidate sparsity matches the ground truth value $k = 50$, under both $(p, q)$ settings.

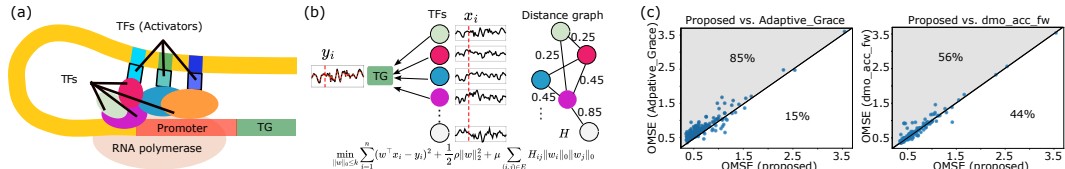

Figure 3: (a) Schematic of transcriptional gene regulation, where transcription factors (TFs) bind to DNA and cooperate with one another to control target gene (TG) expression. (b) Our proposed model for (a) identifies sets of TFs whose expression explains the TG's expression and are close to each other in a protein-protein interaction (PPI) network. (c) Out-of-sample mean squared error (OMSE) comparisons between the proposed model and the top performers (Adaptive_Grace and dmo_acc_fw) in Fig. 2 and S1. Each dot represents a TG, and the dots in the shaded region represent that the proposed model achieves lower OMSE. The percentage is the proportion of dots that fall in that region.

To evaluate robustness against varying and negatively correlated feature weights within clusters, we benchmarked the competing methods using the second random ensemble (Please check **Random Ensemble II** in the supplementary materials). The same conclusion holds in the second random ensemble.

### 4.2 REAL-WORLD APPLICATION: GENE REGULATION INFERENCE

Transcriptional gene regulation in eukaryotes is outlined in Fig. 3a. Transcription factors (TFs) physically bind to DNA to control target gene (TG) expression, while also interacting with one another. Although high-throughput technologies can measure TF and TG expression levels and identify physical TF–TF interactions, they do not directly reveal the complete molecular mechanisms in Fig. 3a. Consequently, leveraging partial observations and domain knowledge to reverse-engineer gene regulation remains a central challenge in systems biology.

We address this challenge by proposing a graph-guided sparse model (see Fig. 3b). Let $y_i$ be the expression level of a single TG in sample $i$, and let $x_i \in \mathbb{R}^d$ be the expression levels of $d$ candidate TFs. The distances between $d$ candidate TF in in the Protein-Protein Interaction (PPI) network are encoded in the distance graph, whose adjacency matrix is $H$. Here we use the diffusion state distance proposed in Cao et al. (2013). Our model seeks to identify TFs that (1) strongly explain the TG's expression patterns and (2) are physically proximate in the PPI network, reflecting the cooperative nature of TF regulation.

We evaluate our method using scMultiome-seq PBMC 3k data 10x Genomics (2021), focusing on paired scRNA-seq and scATAC-seq profiles from CD4 T cells. After standard preprocessing, we analyze 182 highly variable TGs. For each TG, potential TFs are selected via motif matching Bailey et al. (2009) in open chromatin regions, resulting in a set of TF candidates for our inference task. The PPI network we used is downloaded from BioGRID Oughtred et al. (2021). Since ground-truth regulatory interactions are not fully established, we assess performance by out-of-sample mean squared error (OMSE), choosing the top-5 TFs per method. More details about the data used in this experiment are provided in the supplementary materials.

Figure 3c shows the OMSE comparison between the proposed model and Adaptive_Grace and dmo_acc_fw (the top performers in Fig. 2 and S1). The GFL_Proximal method Xin et al. (2014) fails on most TGs by using the source codes provided in Xin et al. (2014), therefore, we exclude it in the comparison. The comparisons between the proposed model and other competing methods and the ablation study are provided in the supplementary materials. From the comparison results, we find that for the majority of the TG, the TFs inferred by the proposed model achieve lower OMSE, indicating outperformance of the proposed model in gene regulation inference.

## 5 CONCLUSION

In this paper, we introduce a novel convex framework for learning structured sparsity by integrating Boolean relaxation techniques into graph-guided sparse learning models. We provide theoretical tools to verify the exactness of the solution of the relaxation, ensuring that the relaxed solution coincides with the optimal integral solution under certain conditions. Additionally, we develop a rounding algorithm to produce a feasible integral solution when the relaxation yields a fractional one. For the case of least-squares loss, we conduct extensive experiments to demonstrate the effectiveness of the proposed framework, highlighting its advantages over traditional methods in terms of both accuracy and computational efficiency.

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

# Supplementary Material

## A    PROOF OF THEOREM 2.1

The initial step of our reformulation aligns closely with Pilanci et al. (2015); Wang et al. (2023), apart from the inclusion of the final term. For completeness, we provide a more detailed derivation here and also correct a few typographical errors from those earlier work.

To begin with, we focus on the first two terms in the original objective function. Let $D(u)$ be a diagonal matrix whose $i$-th diagonal element is $u_i$. Hence, each row of $D(u)$ corresponds to a single feature $i$. So we can do the change of variable $w = D(u)w$ where $u_i$ indicates whether $w_i$ is nonzero or not, that is, $u_i = \|w_i\|_0$. In this way, the first two terms can be written as the optimization problem defined as following:

$$P = \min_{w \in \mathbb{R}^d} \left\{ \sum_{i=1}^{n} f\left(\langle D(u)x_i, w\rangle, y_i\right) + \frac{1}{2}\rho\|w\|_2^2 \right\}.$$

Adapting the arguments from Pilanci et al. (2015); Wang et al. (2023), we write $P$ as the following minimax problem:

$$P = \min_{w \in \mathbb{R}^d} \max_{v \in \mathbb{R}^n} \left\{ \sum_{i=1}^{n} \left(w^\top D(u)\, x_i\right)\, v_i - f^*(v_i, y_i) + \frac{1}{2}\rho\|w\|_2^2 \right\}$$

$$= \max_{v \in \mathbb{R}^n} \min_{w \in \mathbb{R}^d} \left\{ \sum_{i=1}^{n} \left(w^\top D(u)\, x_i\right)\, v_i - f^*(v_i, y_i) + \frac{1}{2}\rho\|w\|_2^2 \right\}.$$

Here, $f^*$ denotes the convex conjugate of $f$. The unique minimizer of the inner problem (with respect to $w$) is

$$w^* = -\frac{1}{\rho} \sum_{i=1}^{n} D(u)\, x_i\, v_i.$$

Substituting $w^*$ back into the objective yields

$$P = \max_{v \in \mathbb{R}^n} \left\{ \sum_{i=1}^{n} \left(-\frac{1}{\rho}\sum_{j=1}^{n} v_j\, x_j^\top D(u)\right) D(u)\, x_i\, v_i - \sum_{i=1}^{n} f^*(v_i, y_i) + \frac{1}{2}\rho\left(\frac{1}{\rho}\sum_{i=1}^{n} v_i\, x_i^\top D(u)\right)^2 \right\}$$

$$= \max_{v \in \mathbb{R}^n} \left\{ -\frac{1}{2\rho}\sum_{i=1}^{n}\sum_{j=1}^{n} v_j\, x_j^\top \left(D(u)D(u)\right) x_i\, v_i - \sum_{i=1}^{n} f^*(v_i, y_i) \right\}$$

$$= \max_{v \in \mathbb{R}^n} \left\{ -\frac{1}{2\rho}\left\|D(u)\, X^\top v\right\|^2 - \sum_{i=1}^{n} f^*(v_i, y_i) \right\}.$$

where $X$ is the matrix whose rows (or columns, depending on convention) correspond to $x_i^\top$.

Combining the above inner-problem result with the graph-guided regularization, we arrive at the final reformulation:

$$P^* = \min_{u \in \{0,1\}^d} \left\{ \max_{v \in \mathbb{R}^n} \left[ -\frac{1}{2\rho}\left\|D(u)\, X^\top v\right\|^2 - \sum_{i=1}^{n} f^*(v_i, y_i) \right] + \mu\,\mathrm{tr}\left(u^\top H u\right) \right\}.$$

When we take $H = L$, the third term in the original problem corresponds to a *minimum binary cut* that separates selected and non-selected features in the graph. Specifically, it quantifies the minimum number of (noisy or incorrect) edges that must be removed to disconnect these two groups. This interpretation follows from Section 4 of Fan & Pardalos (2010) and Section 2 of Fan & Pardalos (2012), which establish that

$$\text{minimum cut} = \frac{1}{2}\,\mathrm{tr}(M^\top L M),$$

where $M$ is the assignment matrix indicating group membership of the nodes. For binary cuts, each node belongs to exactly one of two groups (selected or non-selected), and thus $M \in \mathbb{R}^{d \times 2}$ reduces to a single column vector $u \in \{0,1\}^d$. In this case, the group membership is fully encoded by $u$, eliminating the need for two columns or the symmetry-adjusting factor of $\frac{1}{2}$. Therefore, we obtain:

$$\text{minimum cut} = \frac{1}{2}\operatorname{tr}(M^\top L M) = \operatorname{tr}(u^\top L u) = u^\top L u,$$

where we drop the trace notation since the expression is scalar-valued.

A similar interpretation applies when $H = \widetilde{L}$, the normalized Laplacian. In this case, the corresponding term captures the *normalized cut* associated with the binary partition defined by $u$.

## B    PROOF OF THEOREM 2.2

We establish the exact recovery conditions via convex optimality theory. First, recall the first-order convex optimality condition for constrained minimization:

**Theorem B.1** (Nesterov et al. (2018)). *Suppose $\hat{x}$ is a local minimizer of a differentiable function $F : \mathbb{R}^d \to \mathbb{R}$ on a closed convex set $\mathcal{X} \subseteq \mathbb{R}^d$. If $F$ is differentiable at $\hat{x}$, then*

$$-\nabla F(\hat{x}) \in \mathcal{N}_{\mathcal{X}}(\hat{x}),$$

*where $\mathcal{N}_{\mathcal{X}}(\hat{x})$ is the normal cone of $\mathcal{X}$ at $\hat{x}$.*

In particular, if the feasible set $\mathcal{X}$ is a polyhedron given by linear inequalities, we have the following specific description of its normal cone:

**Theorem B.2** (Nesterov et al. (2018)). *Let $A \in \mathbb{R}^{c \times n}$ and $\beta \in \mathbb{R}^c$. Define the polyhedron $Q(A, \beta) = \{x \mid Ax \le \beta\}$. For any $x \in Q(A, \beta)$, the normal cone at $x$ is*

$$\mathcal{N}_{Q(A,\beta)}(x) = \left\{ A^\top \lambda \,\middle|\, \lambda \in \mathbb{R}^c, \ \lambda \ge 0, \ \lambda^\top(\beta - Ax) = 0 \right\}.$$

Let $\hat{u}$ be the integral optimal solution of the original problem $P^*$. Then $\hat{u}$ is also optimal for the relaxed program $P_{\text{BR}}$ if and only if

$$-\nabla F(\hat{u}) \in \mathcal{N}_{Q(A,\beta)}(\hat{u}),$$

which is equivalent to finding a vector $\lambda \in \mathbb{R}^c$ satisfying

$$\begin{cases} -\nabla F(\hat{u}) = A^\top \lambda, \\ \lambda^\top(\beta - A\hat{u}) = 0. \end{cases} \tag{6}$$

Here $c := 2d + 1$ is the number of constraints in $P_{\text{BR}}$.

The feasible set $\mathcal{X} = Q(A, \beta)$ is the polyhedron defined by

$$A = \begin{bmatrix} I_d \\ -I_d \\ \mathbf{1}_d^\top \end{bmatrix}, \qquad \beta = \begin{bmatrix} \mathbf{1}_d \\ \mathbf{0}_d \\ k \end{bmatrix},$$

where $I_d$ is the $d \times d$ identity matrix. These rows encode

$$\begin{aligned} u_i &\le 1 & (\lambda_i), \\ -u_i &\le 0 & (\lambda_i'), \\ \mathbf{1}_d^\top u &\le k & (\lambda_0). \end{aligned}$$

In other words, the constraints in our relaxed problem can be represented as

$$\underbrace{\begin{bmatrix} 1 & & & & & \\ -1 & & & & & \\ & 1 & & & & \\ & -1 & & & & \\ & & \ddots & & & \\ & & & & 1 & \\ & & & & -1 & \\ 1 & 1 & \cdots & 1 & 1 & 1 \end{bmatrix}}_{A} \underbrace{\begin{bmatrix} u_1 \\ u_2 \\ \vdots \\ \vdots \\ u_d \end{bmatrix}}_{u} \le \underbrace{\begin{bmatrix} 1 \\ 0 \\ 1 \\ 0 \\ \vdots \\ 1 \\ 0 \\ k \end{bmatrix}}_{\beta}$$

which is a polyhedron
$$Q(A, \beta) = \{u \mid Au \le \beta\}.$$

Matrix $A$ thus has $2d + 1$ rows. The first $2d$ rows enforce $0 \le u_i \le 1$ for each $i$, while the last row enforces $\sum_{i=1}^{d} u_i \le k$.

If
$$\hat{v} \in \arg\max \left\{ -\frac{1}{2\rho} v^\top X D(y) X^\top v - \sum_{i=1}^{n} f^*(v_i, y_i) \right\},$$

then we have
$$F(u) := -\frac{1}{2\rho} \hat{v}^\top X D(u) X^\top \hat{v} - \sum_{i=1}^{n} f^*(\hat{v}_i; y_i) + \mu(u^\top H u)$$

Therefore, we have
$$\frac{\partial F}{\partial u_i} = \frac{\partial}{\partial u_i} \left( -\frac{1}{2\rho} \hat{v}^\top X D(u) X^\top \hat{v} \right) + \frac{\partial}{\partial u_i} \left( \mu \cdot (u^\top H u) \right) = -\frac{1}{2\rho} \left( X_i^\top \hat{v} \right)^2 + 2\mu (H u)_i$$

Combining terms:
$$-\nabla F(\hat{u}) = \begin{bmatrix} \frac{1}{2\rho} (X_1^\top \hat{v})^2 - 2\mu(H\hat{u})_1 \\ \vdots \\ \vdots \\ \frac{1}{2\rho} (X_d^\top \hat{v})^2 - 2\mu(H\hat{u})_d \end{bmatrix}.$$

From equation 6, the first equation gives
$$\frac{1}{2\rho} (X_i^\top \hat{v})^2 - 2\mu(H\hat{u})_i = \lambda_i - \lambda_i' + \lambda, \quad i = 1, \dots, d.$$

The second equation is
$$\sum_{i=1}^{d} \lambda_i (1 - \hat{u}_i) + \sum_{i=1}^{d} \lambda_i' \hat{u}_i + \lambda \left( k - \sum_{i=1}^{d} \hat{u}_i \right) = 0.$$

**Case 1:** $i \in I_n$. If $w_i$ is not selected, $\hat{u}_i = 0$. Then $\lambda_i = 0$ and
$$\frac{1}{2\rho} (X_i^\top \hat{v})^2 - 2\mu(H\hat{u})_i = -\lambda_i' + \lambda \le \lambda.$$

**Case 2:** $i \in I_s$. If $w_i$ is selected, $\hat{u}_i = 1$. Then $\lambda_i' = 0$ and
$$\frac{1}{2\rho} (X_i^\top \hat{v})^2 - 2\mu(H\hat{u})_i = \lambda_i + \lambda \ge \lambda.$$

These conditions prove the "only if" direction.

*Proof.* The converse follows constructively: given $\lambda$ satisfying the inequalities, choose
$$\lambda_i = \begin{cases} \frac{1}{2\rho} (X_i^\top \hat{v})^2 - 2\mu(H\hat{u})_i - \lambda, & i \in I_s \\ 0, & \text{otherwise} \end{cases}$$

and
$$\lambda_i' = \begin{cases} \lambda - \frac{1}{2\rho} (X_i^\top \hat{v})^2 + 2\mu(H\hat{u})_i, & i \in I_n \\ 0, & \text{otherwise} \end{cases}$$

which satisfy non-negativity and complementarity. $\qquad\square$

The proof of Corollary 2.3 follows directly that $\hat{v} = -By$.

## C  PROOF OF THEOREM 2.4

Let $u \in \{0,1\}^d$ be a random vector with independent Bernoulli coordinates, and suppose $\mathbb{E}\left[\sum_{j=1}^d u_j\right] \le k$. By Chernoff's bound for Bernoulli sums, for any $\delta > 0$ we have

$$\mathbb{P}\left[\sum_{j=1}^d u_j \ge (1+\delta)k\right] \le c_1 e^{-c_2 k \delta^2},$$

for sufficiently large constants $c_1, c_2$. The Boolean problem admits a saddle-point representation:

$$P^* = \min_{\substack{u \in \{0,1\}^d \\ \sum_{i=1}^d u_i \le k}} \left\{\max_{v \in \mathbb{R}^n} H(u)\right\},$$

where

$$H(u) = \max_{\|v\|_2 \le 2}\left[-\tfrac{1}{2\rho} v^\top X D(u) X^\top v - \|v\|_2^2 - 2v^\top y\right] + \mu \operatorname{tr}(u^\top H u).$$

Since the optimal value is nonnegative, one can choose $v$ with $\|v\|_2 \le 2$. Then

$$\begin{aligned}
H(u) - P^* &\le H(u) - H(\bar{u}) \\
&= \max_{\|v\|_2 \le 2}\left[-\tfrac{1}{2\rho} v^\top X D(u) X^\top v - \|v\|_2^2 - 2v^\top y\right] + \mu \operatorname{tr}(u^\top H u) \\
&\quad - \max_{\|v\|_2 \le 2}\left[-\tfrac{1}{2\rho} v^\top X D(\bar{u}) X^\top v - \|v\|_2^2 - 2v^\top y\right] - \mu \operatorname{tr}(\bar{u}^\top H \bar{u}) \\
&\le \max_{\|v\|_2 \le 2}\left[-\tfrac{1}{2\rho} v^\top X \big(D(u) - D(\bar{u})\big) X^\top v\right] + \mu\big(u^\top H u - \bar{u}^\top H \bar{u}\big) \\
&\le \tfrac{1}{\rho} \sigma_{\max}\Big(X\big(D(u) - D(\bar{u})\big) X^\top\Big) + \mu\big(u^\top H u - \bar{u}^\top H \bar{u}\big),
\end{aligned}$$

where $\sigma_{\max}(\cdot)$ is the maximum eigenvalue of a symmetric matrix. Note that

$$X\big(D(u) - D(\bar{u})\big) X^\top = \sum_{j \in R} (u_j - \mathbb{E}[u_j]) \, X_j X_j^\top \equiv \sum_{j \in R} A_j,$$

where $X_j$ is the $j$-th column of $X$. Each $A_j$ has mean zero and $\|A_j\| \le 1$. By the Ahlswede–Winter inequality Ahlswede & Winter (2002),

$$\mathbb{P}\left[\sigma_{\max}\Big(\sum_{j \in R} A_j\Big) \ge \sqrt{r}\, t\right] \le 2\min\{n, r\} \exp\big(-\tfrac{t^2}{16}\big),$$

where $r$ is the number of fractional components. Setting $t^2 = c \log(\min\{n, r\})$ for a large $c$ ensures the matrix sum is $O(\sqrt{r} + \log n)$ with high probability.

For the second part, Lemma C.1 implies

$$\mathbb{P}\big[\big|u^\top L u - \mathbb{E}[u]^\top L \mathbb{E}[u]\big| \ge t\big] \le 2\exp\Big[-c\min\Big(\tfrac{t^2}{\|L\|_F^2}, \tfrac{t}{\|L\|}\Big)\Big] + 2\exp\Big[-c'\tfrac{t^2}{\|L\bar{u}\|_2^2}\Big].$$

Taking $t^2 = \log(n)$ completes the proof.

**Lemma C.1** (Concentration of $u^\top H u$). *Let $u \in \mathbb{R}^n$ have independent sub-Gaussian coordinates with mean $\bar{u} = \mathbb{E}[u]$ and sub-Gaussian norm bounded by $K$ (take $K = 1$ for Bernoulli). Then for any $H \in \mathbb{R}^{n \times n}$ there are constants $c, c' > 0$ such that for all $t > 0$,*

$$\mathbb{P}\big[\big|u^\top H u - \mathbb{E}[u]^\top H \mathbb{E}[u]\big| \ge t\big] \le 2\exp\Big[-c\min\Big(\tfrac{t^2}{K^4\|H\|_F^2}, \tfrac{t}{K^2\|H\|}\Big)\Big] + 2\exp\Big[-c'\tfrac{t^2}{K^2\|H^\top \bar{u}\|_2^2}\Big].$$

*Proof.* Define $x = u - \bar{u}$ so $\mathbb{E}[x] = 0$. Each $x_i$ is sub-Gaussian and the coordinates are independent. Then

$$u^\top H u = (x + \bar{u})^\top H(x + \bar{u}) = x^\top H x + 2\bar{u}^\top H x + \bar{u}^\top H \bar{u},$$

so

$$u^\top H u - \bar{u}^\top H \bar{u} = x^\top H x - \mathbb{E}[x^\top H x] + 2\bar{u}^\top H x.$$

By the Hanson–Wright inequality, there is $c > 0$ such that

$$\mathbb{P}\big[\big|x^\top Hx - \mathbb{E}[x^\top Hx]\big| \geq t\big] \leq 2\exp\Big[-c\,\min\Big(\tfrac{t^2}{K^4\|H\|_F^2}, \tfrac{t}{K^2\|H\|}\Big)\Big].$$

Since $2\bar{u}^\top Hx = (2H^\top \bar{u})^\top x$ is also sub-Gaussian in $x$, there is $c' > 0$ such that

$$\mathbb{P}\big[\big|2\bar{u}^\top Hx\big| \geq t\big] \leq 2\exp\Big[-c'\,\tfrac{t^2}{K^2\|H^\top \bar{u}\|_2^2}\Big].$$

A union bound on the sum of these two deviations completes the proof. $\qquad\square$

## D  PROOF OF THEOREM 3.1

By definition, if we take $H = L$, we have

$$(Lu)_i = \sum_k L_{ik}u_k = L_{ii}u_i + \sum_{k\neq i} L_{ik}u_k.$$

If $i \in I_n$, then node $i$ is not selected, so $u_i = 0$ and

$$(Lu)_i = \sum_{k\neq i} L_{ik}u_k = - \sum_{k\sim i, k\in I_s} 1,$$

where $k \sim i$ means node $k$ is connected to node $i$ in the graph. In other words, it is the negative of the number of selected nodes connected to the node $i$.

If $i \in I_s$, we have $u_i = 1$, so

$$(Lu)_i = \deg(i) - \sum_{k\neq i, k\in I_s} 1 = \sum_{k\sim i, k\in I_n} 1,$$

where $\deg(i)$ means the degree of node $i$. In other words, it is the number of non-selected nodes connected to the node $i$.

By Corollary 2.3, the exactness holds if and only if one can find the value $\lambda$, such that

$$\frac{1}{2\rho}\big(X_i^\top \hat{v}\big)^2 - 2\mu(Lu)_i \leq \lambda$$

when $i \in I_n$, and

$$\frac{1}{2\rho}\big(X_i^\top \hat{v}\big)^2 - 2\mu(Lu)_i \geq \lambda$$

when $i \in I_s$. Equivalently, for all $i \in I_s$ and $j \in I_n$, we need to find the $\lambda$ such that

$$\frac{1}{2\rho}\big(X_j^\top \hat{v}\big)^2 - 2\mu(Lu)_j \leq \lambda \leq \frac{1}{2\rho}\big(X_i^\top \hat{v}\big)^2 - 2\mu(Lu)_i$$

Let $B := \big(I_n + \rho^{-1}XX^\top\big)$ and we have $\big(X_i^\top \hat{v}\big)^2 = \big(X_i^\top By\big)^2$ in the case of least square of regression.

By definition and $y = Xw + \epsilon$, we have

$$\big(X_i^\top By\big)^2 = \big(X_i^\top BXw + X_i^\top B\epsilon\big)^2.$$

From Lemma D.1 and Lemma D.2 in Wang et al. (2023), if $\rho = n^{1/2+\delta}$, with at least high probability $1 - \big(3k\exp(-c_1 n^{1-2\delta}) + 2d\exp(-n/8) + 2d\exp(-n^{2\delta}/(400\gamma^2 k))\big)$, we have, for every $i \in I_s$,

$$0.8\frac{\rho}{\sqrt{k}} \leq |X_i^\top By| \leq 1.2\frac{\rho}{\sqrt{k}}$$

and for every $j \in I_n$, we have

$$0 \leq |X_j^\top By| \leq 0.2\frac{\rho}{\sqrt{k}}$$

So it is suffices to find the $\lambda$'s to satisfy

$$\frac{1}{2\rho}\big(X_j^\top \hat{v}\big)^2 - 2\mu(Lu)_j \leq \frac{1}{2\rho}\Big(0.2\frac{\rho}{\sqrt{k}}\Big)^2 + 2\mu S \leq \lambda \leq \frac{1}{2\rho}\Big(0.8\frac{\rho}{\sqrt{k}}\Big)^2 - 2\mu N \leq \frac{1}{2\rho}\big(X_i^\top \hat{v}\big)^2 - 2\mu(Lu)_i$$

where $S = \max_{i\in I_n} |\{j \in I_s| \, j \sim i\}|$ and $N = \max_{i\in I_s} |\{j \in I_n| \, j \sim i\}|$.
It follows that if we have $0.02\rho/k + 2\mu S \leq 0.32\rho/k - 2\mu N$, we can choose $\lambda$ in the required interval. Clearly this holds if $\rho \geq 3.4\,k\,2\mu(S + N)$.

Note that $S$ and $N$ can be replaced by any larger upper bounds as needed; for example, one can take the maximal degree if $k$ is unknown.

# E  THEOREM 3.1 WITH NORMALIZED LAPLACIAN

If we take $H = \widetilde{L}$ as the normalized Laplacian, we have similar statement. Similarly,

$$(\widetilde{L}u)_i = \sum_k \widetilde{L}_{ik}u_k = \widetilde{L}_{ii}u_i + \sum_{k \neq i} \widetilde{L}_{ik}u_k.$$

If $i \in I_n$, then node $i$ is not selected, so $u_i = 0$ and

$$(\widetilde{L}u)_i = \sum_{k \neq i} \widetilde{L}_{ik}u_k = - \sum_{k \sim i, k \in I_s} \frac{1}{\sqrt{d_i} \cdot \sqrt{d_k}},$$

where $k \sim i$ means node $k$ is connected to node $i$ in the graph, $d_i$ is the degree of the node $i$. It follows that

$$-2\mu(\widetilde{L}u)_i = 2\mu \sum_{k \sim i, k \in I_s} \frac{1}{\sqrt{d_i} \cdot \sqrt{d_k}} \leq 2\mu \sum_{k \sim i, k \in I_s} \frac{1}{\sqrt{d_i} \cdot 1} \leq 2\mu \frac{S}{\sqrt{S}} \leq 2\mu\sqrt{S}$$

where $S = \max_{i \in I_n} |\{j \in I_s| \, j \sim i\}|$. The maximum is obtained when non-selected node $i$ only connected to selected feature nodes which has degree 1.

If $i \in I_s$, we have $u_i = 1$, so

$$(\widetilde{L}u)_i = 1 - \sum_{k \neq i, k \in I_s} \frac{1}{\sqrt{d_i} \cdot 1}$$

where $d_i = \deg(i)$ means the degree of node $i$. It follows that

$$-2\mu(\widetilde{L}u)_i = 2\mu \left( \sum_{k \neq i, k \in I_s} \frac{1}{\sqrt{d_i} \cdot 1} - 1 \right)$$

$$= 2\mu \left( \sum_{k \neq i, k \in I_s} \frac{1}{\sqrt{d_i}\sqrt{d_k}} - \frac{d_i}{\sqrt{d_i}\sqrt{d_i}} \right)$$

$$= 2\mu \left( \sum_{k \neq i, k \in I_s} \frac{1}{\sqrt{d_i}\sqrt{d_k}} - \sum_{k \neq i, k \in I_s} \frac{1}{\sqrt{d_i}\sqrt{d_i}} - \sum_{k \neq i, k \in I_n} \frac{1}{\sqrt{d_i}\sqrt{d_i}} \right)$$

$$= 2\mu \left( \sum_{k \neq i, k \in I_s} \frac{1}{\sqrt{d_i}} \left( \frac{1}{\sqrt{d_k}} - \frac{1}{\sqrt{d_i}} \right) - \sum_{k \neq i, k \in I_n} \frac{1}{\sqrt{d_i}\sqrt{d_i}} \right)$$

If we decompose $d_i = d_i^s + d_i^n$ into two parts, number of selected node connected to node $i$ and number of non-selected node connected to node $i$, then we have the first part

$$\sum_{k \neq i, k \in I_s} \frac{1}{\sqrt{d_i}} \left( \frac{1}{\sqrt{d_k}} - \frac{1}{\sqrt{d_i}} \right) \geq d_i^s \frac{1}{\sqrt{d_i}} \left( \frac{1}{\sqrt{d_k}} - \frac{1}{\sqrt{d_i}} \right)$$

$$\geq d_i^s \frac{1}{\sqrt{d_i}} \left( - \frac{1}{\sqrt{d_i}} \right)$$

$$\geq - \frac{d_i^s}{d_i}$$

The second part is

$$- \sum_{k \neq i, k \in I_n} \frac{1}{\sqrt{d_i}\sqrt{d_i}} = - \frac{d_i^n}{d_i}$$

So we have

$$-2\mu(\widetilde{L}u)_i \geq 2\mu \left( - \frac{d_i^s}{d_i} - \frac{d_i^n}{d_i} \right) = -2\mu$$

Similar with the proof of Theorem 3.1, it follows that if we have $0.02\rho/k + 2\mu\sqrt{S} \leq 0.32\rho/k - 2\mu$, we can choose $\lambda$ in the required interval. Clearly this holds if $\rho \geq 6.8\,\mu k(\sqrt{S} + 1)$.

# F    OPTIMIZATION

We employ the projected Quasi-Newton (PQN) method to optimize the objective function $P_{\text{BR}}$ as defined in Equation (5). The details of PQN are elaborated in Schmidt et al. (2009), where Algorithm 1 provides the step-by-step procedure. We refer interested readers to Schmidt et al. (2009), for further details. To apply PQN, we require the gradient of the objective function in Equation (5). The partial gradient with respect to $u_i$ is given by

$$\frac{\partial G(u)}{\partial u_i} = -\frac{1}{2\rho} \left( X_i^\top \left( \frac{1}{\rho} X D(u) X^\top + I \right)^{-1} y \right) + 2\mu(Lu)_i. \tag{7}$$

Computing this gradient involves solving a rank-$\|u\|_0$ linear system of size $n$, which can be computed in time $\mathcal{O}(\|u\|_0^3) + \mathcal{O}(nd)$ using QR decomposition. Given that the sparsity level $k$ is relatively small, this computation remains efficient. Additionally, we must perform the following projection step in PQN:

$$\min_{x \in \Omega} \|x - y\|_2^2,$$

where $\Omega$ is defined in Section 2.2. The projection onto the relaxed constraint set $\Omega$ can be efficiently handled by a commercial solver (we use Gurobi Optimization, LLC (2024)). We can also leverage the fast projection algorithm mentioned in the paper.

# G    RUNTIME COMPARISON

We present the computational efficiency of different methods in Table 1. To ensure a fair comparison, we restrict the runtime analysis to the proposed method and the generalized fused lasso (GFL) baseline, as both are implemented in `MATLAB` and executed from Python via the `matlab.engine` interface. The reported runtimes account for all overhead associated with writing input data to disk for MATLAB consumption and retrieving results back into Python. Consequently, comparing these values with those from methods implemented purely in Python—particularly those leveraging native multiprocessing—would be inappropriate. Note that the reported numbers do not include time for parameter tuning or cross-validation.

Despite incorporating a structured sparsity term, our method maintains high efficiency and is even faster than Proposed ($\mu = 0$), even though the latter does not require evaluating the graph regularization term $u^\top L u$. The speedup arises from leveraging graph structure to accelerate convergence within our projected quasi-Newton (PQN) framework. In contrast, GFL_Proximal methods require substantially more iterations to converge, making them approximately seven times slower than our approach. Other algorithms, such as those based on the alternating direction method of multipliers (ADMM) and coordinate descent for $\ell_1$-regularized GFL, are even slower—further emphasizing the computational advantage of our method.

## G.1    RANDOM ENSEMBLE II: VARIATION IN WEIGHTS AND NEGATIVELY CORRELATED FEATURES

To evaluate robustness against varying and negatively correlated feature weights within clusters, we introduce random perturbations drawn from $\mathcal{N}(0, 0.01)$. Concretely, each weight is modified as $w_j \leftarrow w_j + \mathcal{N}(0, 0.01)$ for all $j$, and we additionally flip the signs of a random subset of selected features. These manipulations approximate real-world scenarios in which correlated features may exhibit slight deviations in magnitude or even opposite signs. Since the initial feature weight magnitude is $1/\sqrt{50} \approx 0.141$, the chosen variance of 0.01 corresponds to roughly a 1% perturbation.

We retain the same parameter settings as in Random Ensemble I ($d = 1000$, $k = 50$, $\gamma = 0.5$) and employ the two graph structures ($p = 0.9, q = 0.2$) and ($p = 0.7, q = 0.2$).

Table 1: Computational efficiency comparison across methods ($n = 1000$, averaged over 10 trials). Time is reported in seconds as mean ± standard deviation.

| Metric | Proposed | Proposed ($\mu = 0$) | GFL_Proximal | Lasso | Adaptive Grace |
|---|---|---|---|---|---|
| Time (s) | $7.42 \pm 0.99$ | $8.02 \pm 0.70$ | $51.94 \pm 27.03$ | $0.04 \pm 0.01$ | $126.27 \pm 13.76$ |

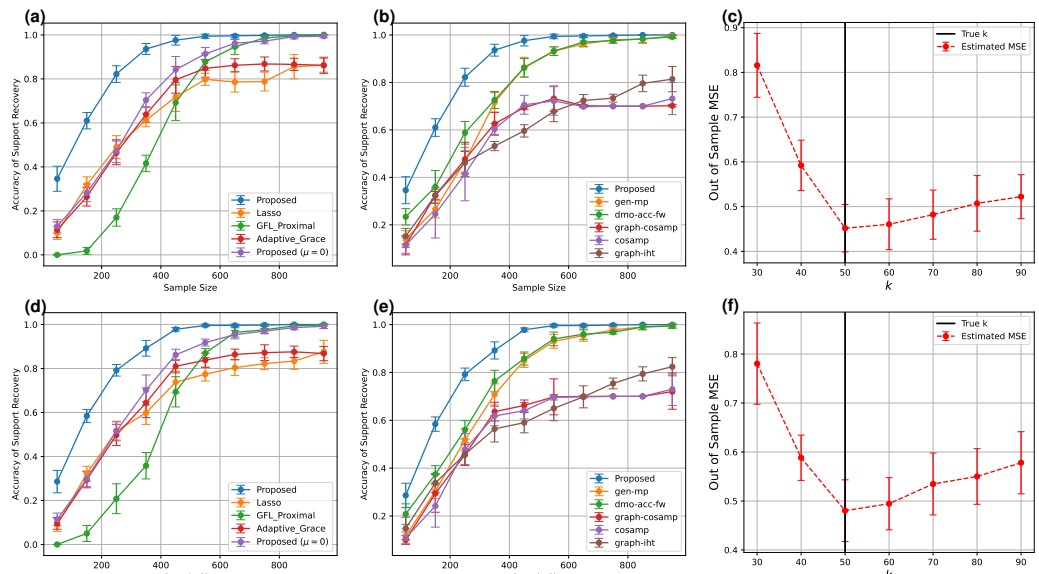

Figure S1: Support recovery performance on **Random Ensemble II**, following the same layout and evaluation protocol as Figure 2 in the main text.

Figures S1 (a), (b), (d), and (e) show that our proposed method consistently achieves high support recovery even at smaller sample sizes, and is the first among all methods to reach perfect recovery (value 1) when $n \approx 500$. In contrast, other methods require significantly more data (around $n \approx 1000$) to approach full recovery. These results highlight the superior robustness of our method in the presence of complex correlation patterns.

In summary, these simulations demonstrate that our approach effectively integrates structural priors with empirical evidence, maintaining strong performance even in high-dimensional and noisy regimes where traditional graph-regularized methods often struggle.

# H   DETAILS OF THE EXPERIMENT OF GENE REGULATION INFERENCE

## H.1   DATA

We use scMultiome-seq PBMC 3k data (10x Genomics, 2021) in our experiments. Specifically, we extract the paired scRNA-seq and scATAC-seq data of CD4 T cells. For the scRNA-seq data, we use regular pipeline (Wolf et al., 2018) to process the data following the legacy workflow provided by Scanpy (`https://scanpy.readthedocs.io/en/stable/tutorials/basics/clustering-2017.html`). For the scATAC-seq data, any peak within 500 bp upstream of a TG's transcription start site is defined as the promoter of the gene, while other open chromatin regions outside of the promoter region but within 250 kb on both sides are defined as distal candidate functional regions. We use FIMO in the MEME suite Bailey et al. (2015) to identify TF candidates that could bind to the open regions detected by the scATAC-seq data.

To run the GRIP model, we also need a PPI network as input. Therefore, we download the PPIs from BioGRID (Oughtred et al., 2019) and extract only the physical interactions with two evidence to construct the PPI network for the GRIP model.

## H.2   DATA SPITING & HYPER-PARAMETER TUNING

We split the data into two parts. The first 80% data is used to train the competing methods. The rest of the 20% data is used to compute the OMSE.

We selected the hyperparameters of the competing methods by cross-validation on the 80% training data.

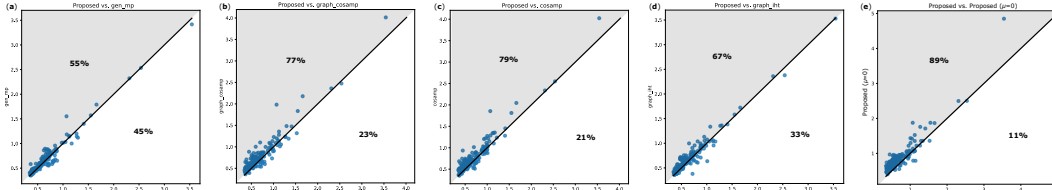

Figure S2: The OMSE comparison result. Each dot represents a TG, and the dots in the shaded region represent that the proposed model achieves lower OMSE. The percentage is the proportion of dots that fall in that region. (a) OMSE comparison between the proposed and gen_mp. (b) OMSE comparison between the proposed and graph_cosamp. (c) OMSE comparison between the proposed and cosamp. (d) OMSE comparison between the proposed and graph_hit. (e) Ablation study: OMSE comparison between proposed and proposed ($\mu = 0$)

### H.3 COMPARISON & ABLATION STUDY

The comparison between the proposed model and the competing methods (gen_mp, graph_cosamp, cosamp, graph_hit) is shown in Fig. S2a-d. The comparison between the proposed model and Adaptive Grace and dmp_acc_fw is shown in the main text Fig. 4. We did the ablation study, which is to compare the proposed model with the proposed model with $\mu = 0$ (as shown in Fig. S2e). All the comparison results in terms of OMSE demonstrate that the proposed model outperforms others.

## I CODE AVAILABILITY

The code for the proposed method, along with all comparison algorithms, data generation, and the random ensemble implementation, can be found here: `https://anonymous.4open.science/r/Graph-Guided-Sparse-Learning-C5D3/`

