# OpenReview forum: "Graph-guided Sparse Learning via Boolean Relaxation"
_ICLR.cc/2026/Conference — ICLR 2026 Conference Withdrawn Submission_

### Official Review · Reviewer_YzwD · 2025-11-01

**Soundness:** 3
**Presentation:** 2
**Contribution:** 3
**Rating:** 4
**Confidence:** 4

**Summary:**

This paper defines a non-convex optimization problem as shown below and proposes a method to solve it by relaxing the Boolean variable $u = (I(w_1 \neq 0), \dots, I(w_d \neq 0))'$ to a continuous interval $[0,1]^d$.
$
\min_{w} \sum_{i=1}^n f(w' x_i, y_i) + \frac{\rho}{2} \||w\||_2^2 + \mu \cdot u'H u \quad\text{subject to}\quad \|| w\||_0 \leq k
$
where $H$ is a positive-semidefinite matrix, which represents the topological properties encoded in a given graph G, e.g. a Laplacian matrix of G. So this problem is a graph-guided sparse learning induced by the L0 penalty.

This paper focuses in particular on the non-convex hard constraint $\|| w\||_0 \leq k$ (L0-norm constraint), and provides a theoretical analysis of the above problem, including conditions under which the relaxed continuous solution matches the true integer solution, as well as probabilistic analysis of an ensemble over random instances. As a concrete example, it examines Least-Squares Regression (specifically, Ridge regression with an L0 penalty), illustrating the corresponding objective function and identifiability conditions, and also reports empirical validations using both artificial random instances and real-world data from gene regulatory networks.

**Strengths:**

- The paper presents a detailed theoretical analysis of the non-convex optimization problem with an L0 penalty, and also proposes practical methods to solve it through convex relaxations using Legendre-Fenchel conjugate and continuous relaxations of integer (boolean) variables.
- Under the assumption that the loss function is convex, the analysis provides several interesting results, including conditions under which the relaxed continuous solution matches the true integer solution, and a probabilistic analysis of the average (ensemble) behavior over random instances.
- Numerical experiments are conducted to empirically evaluate the ensemble behavior on random instances, and the method is also applied to real-world datasets.

**Weaknesses:**

- The proposed formulation is presented in an ad hoc manner, and it remains unclear why or how it constitutes a meaningful problem setup. Rather than evaluating the validity or usefulness of the formulation itself, the paper focuses entirely on technical details of how to solve it, such as whether the adopted relaxation strategies lead to correct integer solutions as defined, or whether they perform well on ensembles of random instances. As a result, the value and justification of the formulation remain unconvincing and unclear. While it's certainly important to find exact solutions to technically hard optimization problems, it seems even more crucial to first evaluate whether the formulation itself is appropriate. After all, the main contribution of this paper is the proposal of a new formulation.

- Throughout the paper, the graph Laplacian is used as an example for ( $H$ ), but in the case of a graph Laplacian, the penalty term in the proposed formulation becomes ( $\sum_{(i,j) \in E} (\||w_i\||_0 - \||w_j\||_0)^2$ ), as noted on p.2, line 106. Essentially, this imposes a strong regularization that encourages both ( $w_i$ ) and ( $w_j$ ) to be nonzero together if there is an edge between them in the given graph ( $G$ ). Intuitively, this can be beneficial if the structure of ( $G$ ) is perfectly accurate and the underlying assumption (that connected pairs should both be nonzero) holds true. However, if we have inaccuracies in the graph (I guess that'll be general), this regularization may strongly mislead the learning process, potentially resulting in an extremely biased solution. The paper does not offer any justification for this formulation, nor does it clarify whether obtaining an exact solution under this setup is meaningful. In practice, structural information in graphs and sparsity constraints are often soft or noisy, so existing methods that allow more flexibility may actually be more appropriate.

- For a study that claims to focus on theoretical aspects, the details are carelessly presented, making it difficult to follow some parts. For example, it is unclear what kind of relationship is assumed between the auxiliary graph ( G ) and the nonzero pattern of the variables. The statement “The matrix ( H ) in equation (2) is a positive semidefinite matrix, which represents the topological properties encoded in ( G )” (line 052) is not sufficient to establish a meaningful connection between the graph structure and the sparsity pattern of the variables. Without a clear assumption or justification linking the topology of ( G ) to which variables should be nonzero, the formulation lacks a coherent foundation.

- Also, norms and absolute values are not clearly distinguished, and the notation makes it hard to tell whether a quantity is a vector or a scalar. While it can be inferred that $w$ is a vector and $w_i$ denotes its scalar components, the use of norm notation even for scalar values raises doubts about the precision and rigor of the definitions. In the most important equation (2), the notation is already confusing. For instance, it uses expressions like ($\|| w_i\||_0$ ), which suggest applying a norm $\|| \cdot ||_0$ to a scalar value $w_i$ though I interpreted this as $I(w_i \neq 0)$. Similarly, the main variable of interest, the Boolean vector ( $u$ ), is written as ( $u = [\||w_1\||_0, \||w_2\||_0, \dots, \||w_d\||_0]^\top$ ), further contributing to the ambiguity. Even in the discussion of baseline methods, there are expressions such as ( $\|| w_i - w_j \||_1$ ) in line 074 and ( $\||w_i - w_j\||_2^2$ ) in line 081, which are difficult to interpret if ( $w_i$ ) and ( $w_j$ ) are scalars. Altogether, the inconsistent and unclear notation makes the paper hard to follow.

**Questions:**

- Could you elaborate a bit more on the justification and motivation behind this main formulation of eq (1)?
- The graph G (or H) is defined quite generally, and the formulation does not explicitly assume any direct relationship between the structure of ( G ) and the sparsity (nonzero) pattern of the variables. Could you clarify this point? If you're using the Laplacian as ( H ), then in general settings, where the provided graph ( G ) may contain errors or noise, this formulation may not be very robust. Do you have any counterarguments to this concern?
- The motivating example in Figure 1 feels highly contrived, as the graph seems to directly encode the correct answer. While it may be a useful illustration in the sense that existing methods fail on it, the setup appears overly tailored to favor the proposed method. If a few random edges were added to the graph, would the proposed approach still be robust and outperform existing methods?
- Regarding the use of norm notation for scalar values: is this simply a notational error, or is this kind of expression standard in this field? At the very least, it’s not common practice, and using it without explanation is extremely confusing. Since this appears even in the definition of the primary variable ( u ), I believe it’s essential to revise this.

---

### Official Review · Reviewer_T4xr · 2025-11-02

**Soundness:** 3
**Presentation:** 3
**Contribution:** 4
**Rating:** 6
**Confidence:** 2

**Summary:**

This paper proposes a novel graph-guided sparse learning framework that penalizes with \ell_0 norm rather than coefficient smoothness. The authors provide an exact Boolean convex reformulation with necessary and sufficient conditions for relaxation and optimality gap. Experiments demonstrate superiority in small-sample regimes on synthetic data and gene regulation inference.

**Strengths:**

- The regularization term encourages selection patterns aligned with minimum/normalized cuts of the graph, avoiding the over-smoothing problem of GFL/Adaptive Grace.
- The theoretical framework provides exact Boolean convex reformulation and verifiable necessary and sufficient conditions for when the continuous relaxation recovers the integral optimum, giving practitioners a certificate of optimality.
- Synthetic experiments show the proposed method is the first to reach perfect recovery around n = 500, while baselines plateau around n=500 excepting some cases of GFL_Proximal. Cross-validation correctly identifies around true k with estimated MSE.
- On gene regulation inference , the method achieves better performance from baselines.

**Weaknesses:**

- While the formulation (Eq. 1) supports general loss functions f, all theoretical guarantees and nearly all experiments focus on squared loss.
- Real-world experiments are limited to a single domain of gene regulation, and experiments with other graph-structured problems are omitted.
- They used a commercial solver, which can raises reproducibility concerns.

**Questions:**

- Can you provide small-scale benchmark results for logistic regression or Huber loss following the general formulation?
- Can you provide some practical scenarios in which a graph contains errors?
- What is the largest d you have tested? Can you report runtime and accuracy at more larger d > 1000?

---

### Official Review · Reviewer_C14w · 2025-11-02

**Soundness:** 3
**Presentation:** 3
**Contribution:** 2
**Rating:** 4
**Confidence:** 4

**Summary:**

This submission proposes a new $\ell_0$-based sparse learning model that integrates graph structure directly into the sparsity constraint rather than enforcing smoothness among neighboring coefficients. The authors reformulate the original nonconvex problem as a Boolean convex program and then introduce a Boolean relaxation with provable conditions for exactness. The authors provide theoretical guarantees of tightness and near-optimal sample complexity. Empirically, the proposed framework significantly outperforms existing graph-guided sparse methods, particularly when the sample size is small. Experiments demonstrate strong results in a real-world gene regulation inference task, highlighting the model’s practical applicability.

**Strengths:**

Three strong points are as follows:

1. The paper provides an exact Boolean reformulation of the $\ell_0$-constrained graph-guided learning problem and establishes necessary and sufficient conditions for the Boolean relaxation to be tight. The authors provide a theoretical analysis that further includes high-probability guarantees of exactness and near-optimal sample complexity on random ensembles.

2. Unlike existing graph-regularized methods, the proposed model leverages the graph to shape the support structure of selected features. This allows flexible coefficient variation while capturing meaningful topological relations, offering a distinct and interpretable inductive bias.

3. The authors demonstrate consistent empirical gains across synthetic and real datasets, particularly under small-sample regimes. The application to gene regulation inference provides a convincing real-world example where the proposed approach outperforms strong baselines, supporting both its practical relevance and robustness.

**Weaknesses:**

1. The submission has a good theoretical analysis. However, I suspect its applicability to specific domain areas is very limited. The topic is a bit outdated compared to modern graph analysis methods (e.g., GNNs or LLMs). Could the authors comment on how the proposed methods compare with the GNN models?

2. Although the Boolean relaxation is theoretically elegant, the proposed algorithm still involves solving convex programs with high computational cost (scaling roughly as $O(\min(n, d)^3))$. The paper lacks large-scale experiments to demonstrate practical feasibility on high-dimensional datasets.

**Questions:**

Please see Weaknesses.

---

### Note · Authors · 2025-12-11

I have read and agree with the venue's withdrawal policy on behalf of myself and my co-authors.